# A mechanistic model of the BLADE platform predicts performance characteristics of 256 different synthetic DNA recombination circuits

Jack E. Bowyer[1,2], Chloe Ding[3,4], Benjamin H. Weinberg[5], Wilson W. Wong[3,4], Declan G. Bates[1,2]*

1 School of Engineering, University of Warwick, Coventry, United Kingdom, 2 Warwick Integrative Synthetic Biology Centre, University of Warwick, Coventry, United Kingdom, 3 Department of Biomedical Engineering, Boston University, Boston, Massachusetts, United States of America, 4 Biological Design Center, Boston University, Boston, Massachusetts, United States of America, 5 Department of Genetics, Harvard Medical School, Boston, Massachusetts, United States of America

* d.bates@warwick.ac.uk

**Data Availability Statement:** All relevant data are within the manuscript and its Supporting Information files.

## Abstract

Boolean logic and arithmetic through DNA excision (BLADE) is a recently developed platform for implementing inducible and logical control over gene expression in mammalian cells, which has the potential to revolutionise cell engineering for therapeutic applications. This 2-input 2-output platform can implement 256 different logical circuits that exploit the specificity and stability of DNA recombination. Here, we develop the first mechanistic mathematical model of the 2-input BLADE platform based on Cre- and Flp-mediated DNA excision. After calibrating the model on experimental data from two circuits, we demonstrate close agreement between model outputs and data on the other 111 circuits that have so far been experimentally constructed using the 2-input BLADE platform. Model simulations of the remaining 143 circuits that have yet to be tested experimentally predict excellent performance of the 2-input BLADE platform across the range of possible circuits. Circuits from both the tested and untested subsets that perform less well consist of a disproportionally high number of STOP sequences. Model predictions suggested that circuit performance declines with a decrease in recombinase expression and new experimental data was generated that confirms this relationship.

## Author summary

A major objective in synthetic biology is to predictably design and construct genetic circuits to control cellular functions. Although recent years have seen numerous advances towards this goal, Synthetic Biology is still mostly a microbial-centric discipline, and high performance genetic circuits are currently lacking in mammalian cells. Site-specific DNA recombinases Cre, Flp are among the most powerful genome engineering tools and form the basis of Boolean logic and arithmetic through DNA excision (BLADE), a platform that

**Funding:** Research supported by funding from a Bilateral BBSRC/NSF award (BB/P011926/1 /#1614642 BIO-Rewritable biocomputers in mammalian cells) to DGB and WWW.bbsrc.ukri. orgnsf.gov The funders had no role in study design, data collection and analysis, decision to publish, or preparation of the manuscript.

**Competing interests:** The authors have declared that no competing interests exist.

has the potential to revolutionise cell engineering for therapeutic applications in mammalian cells. Here, we develop the first mechanistic mathematical model of the 2-input BLADE platform and apply it by simulating the performance of 113 different circuits that have been constructed and tested experimentally. We demonstrate the predictive power of our model by simulating the performance of the 143 circuits that are yet to be tested. Our model is also capable of testing experimental hypotheses, revealing that the performance of synthetic BLADE circuits is sensitive to recombinase expression levels. We were able to confirm this computational result by generating new experimental data.

## Introduction

A fundamental goal of synthetic biology is to reliably design and construct genetic circuits capable of controlling cellular functions [1–4]. Although considerable progress has been made towards this goal in recent years, synthetic genetic circuits in eukaryotic cells have not yet reached the sophistication of their prokaryotic counterparts [5,6]. In addition, the challenge of constructing novel genetic circuits *in vivo* is complicated by the necessity for layering of multiple genetic modules and components [7]. The Boolean logic and arithmetic through DNA excision (BLADE) platform was developed in order to address these challenges, and can be used to construct Boolean logic circuitry reliably in mammalian cells whilst integrating circuit signals on a single transcriptional layer [8]. These circuits are Boolean in the sense that they provide outputs in response to TRUE/FALSE input conditions however, unlike conventional logic gate circuits, once a state has been induced it is fixed and cannot be altered via subsequent input alterations. In the 2-input BLADE platform, up to four distinct cellular states are available through two orthogonal DNA recombination reactions that rewrite genetic sequences and thus provide a lasting effect as the cell divides. This memory feature is particularly beneficial as it allows genetic functions to be turned 'on' or 'off' without the requirement of sustained or continuous stimulation [9–13]. The BLADE platform offers a robust and convenient framework for producing multi-input-multi-output circuits. These circuits can be used for tissue engineering applications, whereby different genes can be specifically expressed in different tissues. Similarly, BLADE is particularly useful in advanced animal model development where, again, one can express different genes in different tissues. For instance, it is possible to express two different inducible kill switches in order to kill off different cell types at different times to analyse their impact on development. Finally, recombinase-based circuits can be used in immune cells for cancer diagnostics. When combined with tumour-specific sensors such as synNotch receptors, these circuits can provide combinatorial detection of antigens on cancer cells for highly specific tumour detection based on multiple tumour markers. The BLADE platform can be used to construct circuits describing any of the 256 ($4^4$) possible permutations of four states (states can be repeated). For example the AND (FALSE FALSE FALSE TRUE), OR (FALSE TRUE TRUE TRUE), XOR (FALSE TRUE TRUE FALSE) and NOT (TRUE FALSE) gates that form the basis of logic operations.

DNA recombination is a highly efficient and reliable method for manipulating genetic sequences in both bacterial and mammalian cells [14,15], and has been used to develop a wide-ranging variety of circuitry in synthetic biology, including genetic switches [16] and logic gates [17,18]. DNA recombinases are specialised proteins that are able to invert, delete or insert sections of DNA flanked by specific attachment sites [19]. There are two main groups of recombinases, serine recombinases and tyrosine recombinases. Serine recombinases catalyse recombination reactions that are irreversible in the absence of a recombination directionality

factor (RDF). When bound together with an appropriate RDF however, the reverse reaction is initiated and the original state of the genetic sequence is restored [20]. Serine recombinases typically bind attB and attP attachment sites, performing double-stranded breaks in the DNA. The resultant pseudo-attachment sites are referred to as attL and attR [21–27]. Similarly, the functionality of the tyrosine recombinase λ integrase is dependent upon cell-specific cofactors [28] and thus presents similar issues as the serine recombinases in the development of novel genetic circuitry, due to the complexity of multiple system inputs. In contrast, some tyrosine recombinases, such as Cre and Flp, are capable of reversible DNA recombination without additional cofactors [15,29–31]. These recombinases bind to loxP and FRT attachment sites, respectively, and carry out a series of single-strand exchanges known as a Holliday junction [32]. The attachment sites are unchanged by the process and hence successive recombination reactions can be mediated by the same recombinase [33].

DNA recombination events facilitate three distinct alterations to a genetic sequence. The first is inversion, whereby attachment site pairs arranged in an antiparallel manner (facing each other) are cut and re-ligated at opposite ends causing a 180˚ rotation of the intermediate genetic sequence. Secondly, excision of a genetic sequence occurs when the attachment sites are arranged in a parallel manner (facing the same direction). In this case, the exposed ends of intermediate sequence are re-ligated to form a circle of DNA which dissociates from the original strand. The gap left in the strand is closed up as the exposed ends are brought together and re-ligated. The third alteration, insertion, is the reversal of an excision reaction, whereby the circle of DNA is inserted (or re-inserted) between the two exposed ends of the strand [20]. Many different DNA recombinases can function in a highly site-specific and orthogonal manner, making them well suited to performing stable genetic alterations that can facilitate cellular Boolean logic operations [34].

Recombinase-based genetic switches and logic gates typically consist of one or more recombinase inputs that target a specific arrangement of attachment site pairs in order to elicit the desired output, usually expression of a gene of interest (GOI). The number of distinct logic functions available increases with the number of inputs. However, this is limited by the dynamical complexity and increased burden on cellular resources caused by increasing the number of biological components [3,35]. Furthermore, there are a finite number of recombinases that have been characterised that can be used reliably as inputs [36], which limits scalability regardless of the difficulty associated with the number of components.

The aforementioned distinctions between the functional properties of serine and tyrosine recombinases are a key issue when designing logic circuits. Serine recombinases elicit stable inversion and excision reactions that constitute an effective 'on' switch (Fig 1A and 1B) however, the 'off' switch presents reliability issues. That is, either transitioning the circuit between 'off' and 'on' states is compromised by the lasting effects of the RDF, or the circuit is not viable for therapeutic applications due to the necessity of constant drug induction [37]. Consequently, serine recombinases have been used to design circuitry that exploits unidirectional functions and therefore does not involve transitions back to prior states [18]. Tyrosine recombinases elicit unstable inversion reactions that are not suitable for the design of logic circuitry due to unstable and continuous reversibility (Fig 1C). However, the excision reaction is considered to be stable due to the dissociation of the DNA circle from the original strand, which significantly decreases the likelihood of naturally occurring insertion reactions, and therefore negates the reversibility (Fig 1D). In addition, tyrosine recombinases Cre and Flp have been found to perform reliably in mammalian cells, and hence they are most suited to mammalian cellular logic circuitry such as the BLADE platform.

The analysis of synthetic biological systems through mathematical modelling approaches has become commonplace in the literature [1,2,12,18,29,38] as it offers a highly efficient

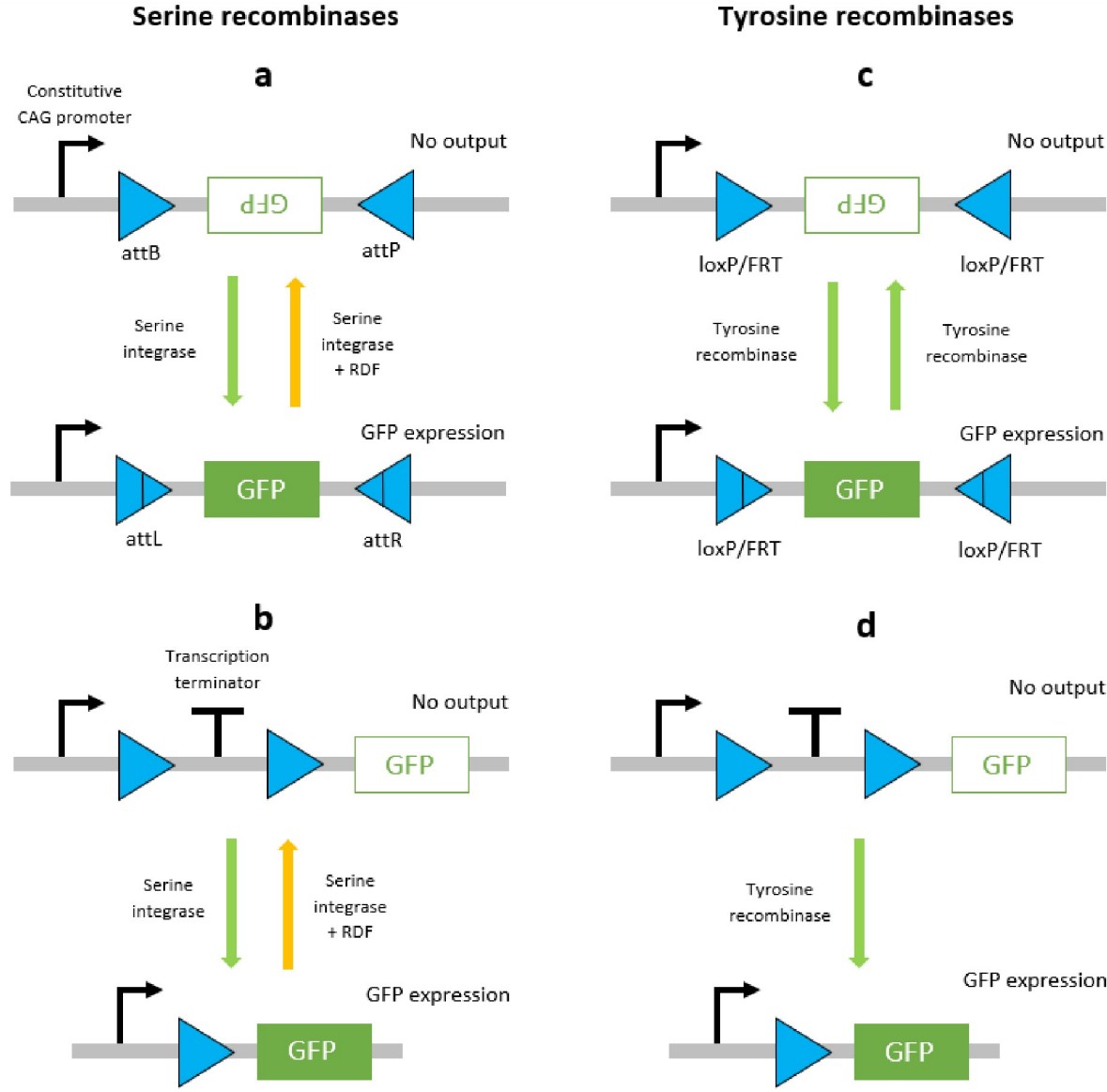

**Fig 1. Schematic diagrams of DNA recombination events mediated by serine and tyrosine recombinases.** Serine integrase-mediated inversion (a) and excision (b) events are stable due to the RDF required to facilitate secondary events. Tyrosine recombinase-mediated inversion (c) and excision (d) events are unstable since the same recombinase can mediate secondary events. The orientation of attachment site pairs gives rise to inversion (antiparallel) or excision (parallel) for both recombinases.

method of examining the behaviour of a system subject to the perturbation of an array of different variables and parameters. These results can provide answers to experimental hypotheses and hence inform the planning and enactment of subsequent experimental studies, saving time and resources. The dissemination of mathematical modelling approaches is important in establishing a wide-ranging archive of useful tools for the design and implementation of novel circuitry across synthetic biology [39–44]. In this paper, we present the first mechanistic mathematical model of the 2-input BLADE platform developed in [8]. The model is derived through the application of mass action kinetics to the Cre- and Flp-mediated recombination reactions that constitute this BLADE platform. The model is calibrated through stochastic Gillespie algorithm simulations in conjunction with a global optimisation algorithm. Our

calibrated model is used to examine trends relating to the best and worst performing circuits. We also predict the effect of decreased recombinase expression level on circuit performance. New experimental data was generated to validate our model predictions, confirming its usefulness as a circuit design tool.

## Results and discussion

### A mechanistic model of the BLADE platform

Our 2-input BLADE model is based on the structure of an existing model of a tyrosine recombinase-based flip-excision (FLEx) switch [45]. The model is comprised of one excision reaction per state transition, with the exception of the transition from $Z_{00}$ to $Z_{10}$, which involves two pairs of Cre-specific loxP attachment sites and hence two excision reactions. The transition from $Z_{00}$ to $Z_{11}$ therefore requires three excision reactions via $Z_{10}$, or two excision reactions via $Z_{01}$. S1 Fig details the network of molecular interactions that form the basis of the 2-input BLADE model. The level of expression of Cre and Flp recombinases in the system is modelled by the parameter $\alpha$. Recombinase (protein) degradation is described by the parameter $\beta_p$. It is assumed that both recombinases are expressed and degraded in the same way, and hence the same parameters are used to describe equivalent reactions. Cre (red circles) and Flp (blue circles) monomers bind to loxP (red rectangles and triangles) and FRT (blue triangles) attachment sites respectively. Recombinase monomer binding is cooperative, meaning that monomers have greater affinity for sites occupied by other monomers. All DNA:protein interactions that occur in the nucleus are modelled as reversible reactions and are each denoted by the corresponding numbered parameter ($k_n$). Holliday junctions consist of five intermediate complexes describing the individual strand exchange reactions that occur during excision [15]. The dilution and dissociation of excised DNA circles out of the system is denoted by the parameter $\delta$. Fig 2 provides a truth table summary of the 2-input BLADE logic operations.

The molecular interactions that comprise the network in S1 Fig represent a system of biochemical equations that can be found in S1 Table. We derive our deterministic mathematical model through the application of mass action kinetics to these biochemical equations. The result is a system of ordinary differential equations (ODEs), each describing the rate of change in concentration of the corresponding molecular entity with respect to time (S2 Table). Details of how the mechanistic model can be used to simulate the transitioning of the BLADE platform between the four addresses can be found in the S1 Text.

### Formulating a metric for evaluating circuit performance

The data generated in [8] record the percentage of cells expressing green fluorescent protein (GFP) and/or mCherry (a member of the monomeric red fluorescent proteins, mRFP) reporter for 113 distinct circuits, subject to their four possible inputs. The percentage of cells expressing fluorescent protein is referred to as the percentage of cells ON. Each of the four addresses is occupied by one of four genes coding for either no fluorescent protein (STOP), GFP, mCherry or GFP and mCherry simultaneously (GFPmCherry). Hence, each input can potentially facilitate expression of neither, either or both reporters, which gives a total of eight GFP/mCherry outputs for each circuit (two outputs per input). Ideally, each gene coding for fluorescence would be expressed by 100% of cells and each STOP sequence would result in 0% fluorescence. Hence, the performance of each circuit can be measured against its expected ideal output, or truth table. For example, the observed performance of Circuit 1 from Table 1 has a total error of 398% across all eight expected outputs, an average of ~50% per output. Circuit 2 has a total error of ~13% at an average of ~2% per output. This indicates that circuit 2 performed significantly better in practice than circuit 1, an observation that is confirmed by

**Fig 2. The initial setup of the 2-input BLADE platform and the four states accessed via each of the four distinct inputs.** In our BLADE model, all transitions between states are described by (stable) excision recombination events mediated by Cre and/or Flp. Each logical 2-digit input relating strictly to the Cre Flp ordered pair corresponds to the BLADE address with the matching subscript.

the scores received by each circuit (11.58° and 1.29° respectively) based on the angular metric which calculates the angle between the 8-dimensional output vector recorded experimentally and its corresponding truth table vector. That is,

$$\theta = arccos\left(\frac{v_e \cdot v_i}{|v_e||v_i|}\right)$$

where $\theta$ is the angular metric in degrees and $v_e$, $v_i$ are the experimental output and ideal truth table vectors respectively [8]. Therefore, the performance of each circuit is measured on a scale of 0–90° where 0° represents optimal performance (the observed vector is identical to the truth table vector) and 90° represents the worst possible performance (the observed vector is the opposite of the truth table vector).

Analysis of the overall trend in observed circuit performance compared to the corresponding ideal truth table outputs reveals that the angular metric can be improved to give a more insightful assessment of circuit performance. The distributions of the data for the percentage of cells ON for fluorescence outputs and non-fluorescence outputs reveal that the means are 66.09% and 1.83% respectively, with standard deviations 13.86% and 4.38% respectively (Fig 3). That is, in cases where the expected outputs are fluorescent protein expression (GFP, mCherry or GFPmCherry) the mean error recorded between the observed percentage of cells

**Table 1. Circuits 1 and 2, corresponding to circuits 1 and 112 from [8] respectively.** Experimental and ideal outputs are given as 8-dimensional vectors; the angular metric gives the angle in degrees between these two vectors. Our adapted metric divides this angle by the number of vector entries that are expected to produce fluorescence i.e. the number of 100s in the ideal output vector.

| Circuit | Observed output (%) | Ideal output (%) | Total error (%) | Average error (%) | $\theta$ (°) | $\bar{\theta}$ (°) |
|---------|---------------------|-------------------|-----------------|-------------------|--------------|--------------------|
| 1 (1)   | [42 53 34 46 44 59 55 69] | [100 100 100 100 100 100 100 100] | 397.97 | 49.75 | 11.58 | 1.45 |
| 2 (112) | [0 0 0 2 0 0 0 89] | [0 0 0 0 0 0 0 100] | 13.22 | 1.63 | 1.29 | 1.29 |

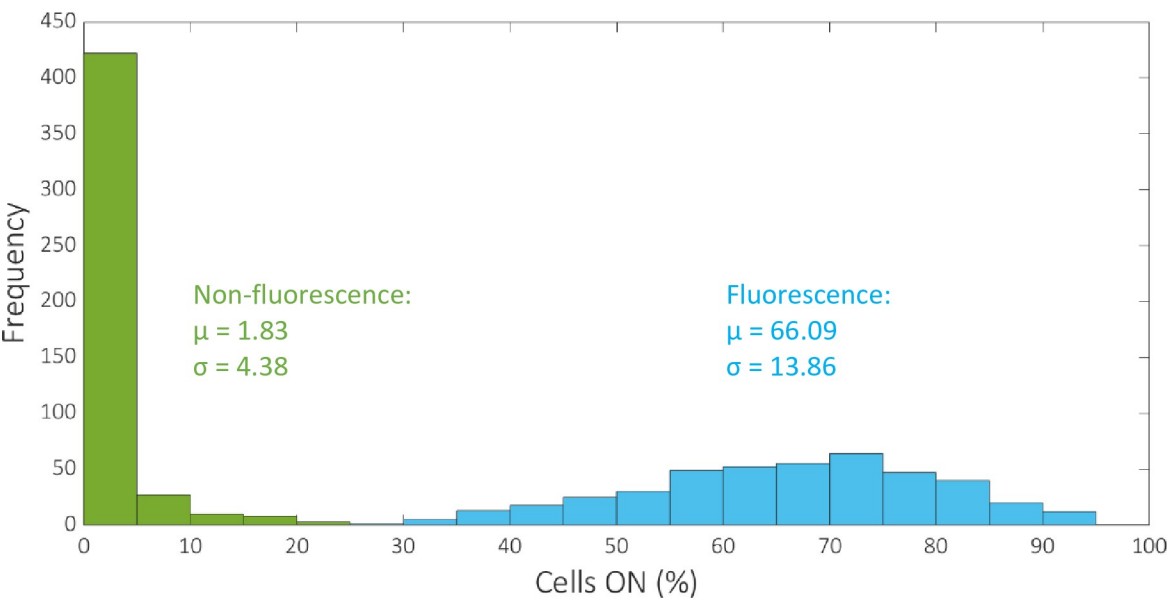

**Fig 3. Histogram showing the distribution of the percentage of cells ON for fluorescence and non-fluorescence outputs.** The mean and standard deviation of the two datasets are denoted by μ and σ respectively. The total number of data values for fluorescence and non-fluorescence outputs are 431 and 473 respectively.

expressing the appropriate output and the corresponding ideal output (100% expression) is ~34%. This figure falls considerably to ~2% error when the outputs in question are the STOP sequences that ideally do not produce any observable fluorescent output (0% expression). Hence, the data demonstrate that a circuit has more difficulty achieving 100% of cells expressing a gene coding for fluorescence than it does achieving 0% of cells expressing fluorescence for a STOP sequence. Furthermore, the angular metric rewards consistency of percentage expression across expected fluorescence outputs, over the absolute magnitude of percentage expression. For this reason we adapt the angular metric to account for the number of outputs that are expected to produce fluorescence. To do this we simply divide the result of the angular metric by the number of expected non-zero outputs for each circuit:

$$\bar{\theta} = \frac{\theta}{n}$$

where n is the number of outputs expected to produce fluorescence and $\bar{\theta}, \theta$ are the adapted and original angular metrics, respectively. Therefore, the original angular metric score for circuit 1 is divided by eight, since all eight outputs are expected to produce fluorescence, and the score for circuit 2 is divided by one, as only one of the outputs is expected to produce fluorescence. This results in the adapted angular metric scores of 1.45 and 1.29 for circuits 1 and 2 respectively and indicates that, although circuit 2 still receives the better score, the performance of the two circuits is not as disparate as the original angular metric suggests. The effect of applying the adapted angular metric to our original dataset is shown in Fig 4.

### Distributions of ensemble simulations reveal consistent performance across the majority of circuits

Given that our angular metric data record the percentage of cells ON across a population, we consider a stochastic modelling approach through the implementation of our model using the Gillespie algorithm. This allows us to perform multiple random simulations of any or all of the

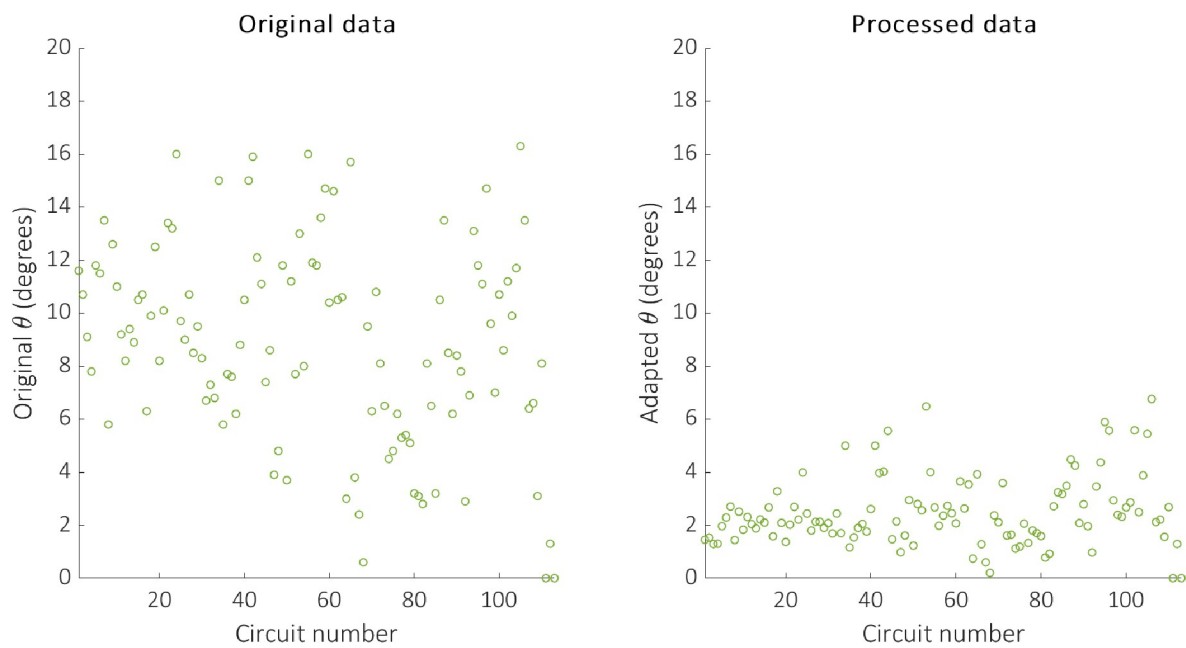

**Fig 4.** The original data generated in [8] (left) and the corresponding processed data based on our adapted angular metric (right).

2-input BLADE circuits and investigate the trends associated with the responses. The Gillespie algorithm facilitates stochastic model simulations without the requirement of formulating stochastic differential equations (SDEs) [46,47]. Running large numbers of stochastic simulations causes a significant increase in computational workload, hence, it is necessary to base the Gillespie simulations on a simplified model of the BLADE platform which requires fewer system states to update at each time point. Our simplified model accounts for the same constitutive expression of Cre and Flp depicted in S1 Fig, but has only four other state variables that correspond to each of the four addresses. Transitioning between states is described by four reversible excision reactions (Fig 5). Applying mass action kinetics to this simplified reaction network derives the following system of model ODEs:

$$\frac{dC}{dt} = \alpha - \beta_p C - k_{1c} Z_{00} C + k_{-1c} Z_{10} Z_{10X} - k_{2c} Z_{01} C + k_{-2c} Z_{11} Z_{11Xc}, \tag{1}$$

$$\frac{dF}{dt} = \alpha - \beta_p F - k_{1f} Z_{00} F + k_{-1f} Z_{01} Z_{01X} - k_{2f} Z_{10} F + k_{-2f} Z_{11} Z_{11Xf}, \tag{2}$$

$$\frac{dZ_{00}}{dt} = k_{-1c} Z_{10} Z_{10X} - k_{1c} Z_{00} C + k_{-1f} Z_{01} Z_{01X} - k_{1f} Z_{00} F - \delta_D Z_{00}, \tag{3}$$

$$\frac{dZ_{10}}{dt} = k_{1c} Z_{00} C - k_{-1c} Z_{10} Z_{10X} - k_{2f} Z_{10} F + k_{-2f} Z_{11} Z_{11Xf}, \tag{4}$$

$$\frac{dZ_{10X}}{dt} = k_{1c} Z_{00} C - k_{-1c} Z_{10} Z_{10X} - \delta_X Z_{10X}, \tag{5}$$

$$\frac{dZ_{01}}{dt} = k_{1f} Z_{00} F - k_{-1f} Z_{01} Z_{01X} - k_{2c} Z_{01} C + k_{-2c} Z_{11} Z_{11Xc}, \tag{6}$$

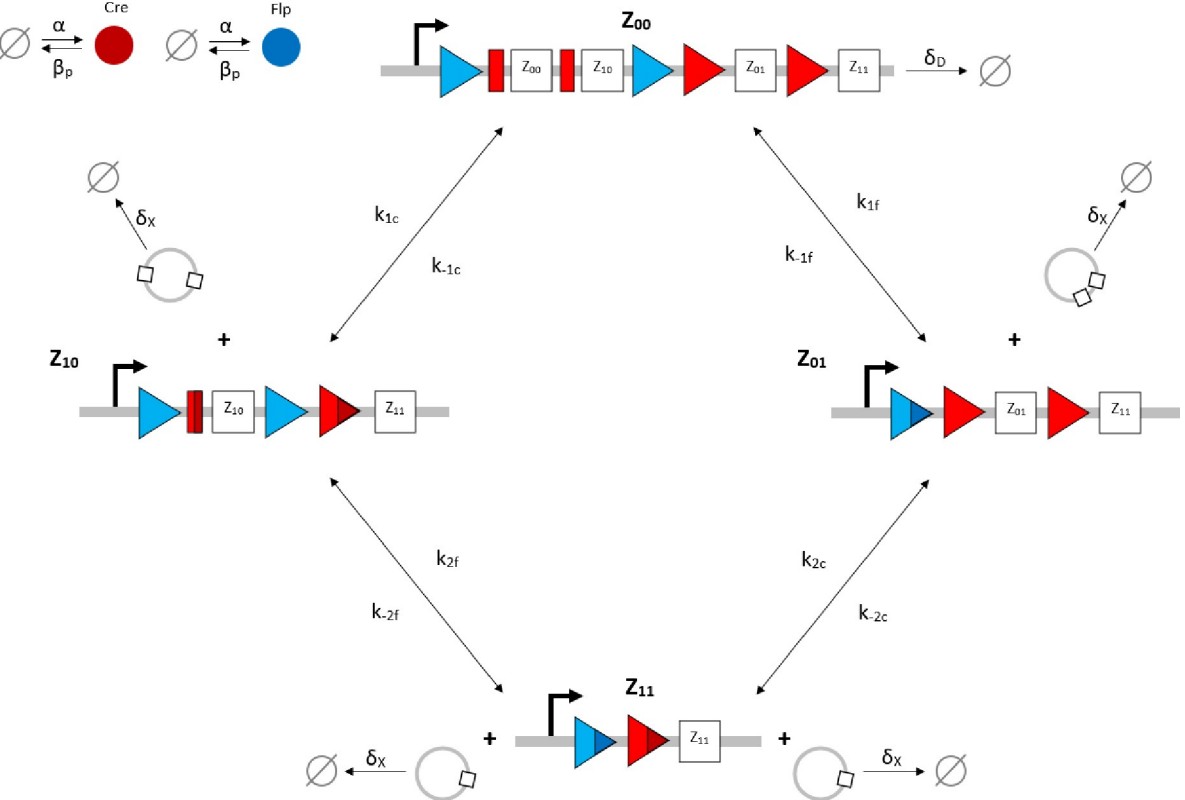

**Fig 5. Schematic diagram of the reaction network that describes the basis of our simplified BLADE model.** The simplified network consists of the minimum number of state variables and parameters in order to minimise the computational workload of running Gillespie algorithm simulations of this simplified model. Tyrosine recombinases Cre (red circles) and Flp (blue circles) are expressed constitutively at the same rate, α. LoxP sites are depicted as red triangles with different shades used to illustrate the result of recombination events. FRT sites are depicted as blue triangles with different shades used to illustrate the result of recombination events. White boxes depict each of the four BLADE addresses. Black arrows depict DNA:protein binding reactions comprising stable excision events. The rate of degradation of recombinase protein is denoted by $\beta_p$. The rate of a Cre monomer binding reversibly to free loxP sites is denoted by $k_{1c}$, $k_{-1c}$ and $k_{2c}$, $k_{-2c}$ for the first and second excision events respectively; the rate of a Flp monomer binding reversibly to free FRT sites is denoted by $k_{1f}$, $k_{-1f}$ and $k_{2f}$, $k_{-2f}$ for the first and second excision events respectively. The rate of dilution of DNA and excised DNA due to cell division is denoted by $\delta_D$ and $\delta_X$, respectively. The empty set symbol is used to depict expression and degradation reactions.

$$\frac{dZ_{01X}}{dt} = k_{1f}Z_{00}F - k_{-1f}Z_{01}Z_{01X} - \delta_X Z_{01X},$$ (7)

$$\frac{dZ_{11}}{dt} = k_{2f}Z_{10}F - k_{-2f}Z_{11}Z_{11Xf} + k_{2c}Z_{01}C - k_{-2c}Z_{11}Z_{11Xc},$$ (8)

$$\frac{dZ_{11Xc}}{dt} = k_{2c}Z_{01}C - k_{-2c}Z_{11}Z_{11Xc} - \delta_X Z_{11Xc},$$ (9)

$$\frac{dZ_{11Xf}}{dt} = k_{2f}Z_{10}F - k_{-2f}Z_{11}Z_{11Xf} - \delta_X Z_{11Xf},$$ (10)

where C and F denote Cre and Flp recombinases respectively; Z denotes BLADE platform states (addresses) corresponding to each subscript; subscripts containing X denote excised DNA with recombinase specificity (c, f) where necessary; α denotes the constitutive expression

of Cre and Flp; $\beta_p$ denotes protein degradation; $\delta_X$ and $\delta_D$ denote the dilution of excised DNA and DNA respectively; reaction rate constants are denoted by numbered k parameters with recombinase specificity where necessary. System variables and parameters are illustrated in the same way, with the exception of the excision reaction rates that are denoted according to each of the two reactions performed by each recombinase ($k_{1c}$/$k_{-1c}$, $k_{2c}$/$k_{-2c}$ and $k_{1f}$/$k_{-1f}$, $k_{2f}$/$k_{-2f}$). The structure of our simplified model network does not retain the mechanistic properties of cooperative monomer binding of the recombinases Cre and Flp. The simplified model also does not retain the two-step Cre-mediated reaction that facilitates the $Z_{00}$ to $Z_{10}$ state transition however, to rectify this, the number of Cre monomers required to elicit the transition is double the number required for all other transitions within the Gillespie algorithm simulations. Note that henceforth all results are generated using the simplified model (Fig 5) and the Gillespie algorithm simulations that it is subjected to.

We utilise global optimisation via a genetic algorithm (GA) in order to parameterise our stochastic model (Materials and Methods). Such algorithms have been widely used in mathematical model inference problems relating to, for example, synthetic oscillators and gene regulatory networks [48–50]. We optimise the model against the adapted angular metric of two circuits that, together, present the full range of possible dynamical responses in equal measure ([GFPmCherry GFP STOP GFP] and [GFPmCherry STOP mCherry mCherry], Table 2). Since we are optimising Gillespie simulations of the model, it is necessary to run as many simulations as is computationally feasible in order to maximise the accuracy of our GA objective function evaluations. That is, stochastic simulations using the same parameter set can provide distinct results, and simulations using distinct parameter sets can provide the same, or very similar, results. This highlights the added importance of the proposed simplification of the model dimensionality/structure, since the genetic algorithm is most effective when the population size is large. We therefore require a large ensemble of simulations for every individual parameter set in a large population of potential solutions, which is computationally intensive. Our objective function must not only identify parameterisations that minimise the error between simulations and the adapted angular metric data across an entire ensemble, but must also minimise the spread (variance) of these errors since failing to do so risks identifying parameterisations that are prone to producing outliers that significantly skew the output of the model. Hence, our objective function takes the mean adapted angular metric across the ensemble of simulations for each circuit and calculates the absolute value of the difference between these means and the experimental data. These results are summed along with the standard deviations of the ensemble simulations for each circuit to provide an overall score to be minimised by the GA (Materials and Methods).

We optimised the model against the adapted angular metric data for each circuit in Table 2 simultaneously in order to determine a single parameterisation capable of providing accurate simulations of their performance. The optimal parameter set identified by the GA can be found in Table 3. We examine the robustness of our optimal parameterisation through multiple runs; the results of which can be found in the S4 Fig. The distribution of the adapted

**Table 2. Circuits 3 and 4, corresponding to circuits 12 and 14 from [8] respectively.** Experimental and ideal outputs are given as 8-dimensional vectors; the angular metric gives the angle in degrees between these two vectors. Our adapted metric divides this angle by the number of vector entries that are expected to produce fluorescence i.e. the number of 100s in the ideal output vector.

| Circuit | Observed output (%) | Ideal output (%) | Total error (%) | Average error (%) | $\theta$ (°) | $\bar{\theta}$ (°) |
|---------|---------------------|------------------|-----------------|-------------------|--------------|---------------------|
| 3 (12)  | [67 76 60 2 6 13 77 0] | [100 100 100 0 0 0 100 0] | 100.38 | 12.55 | 8.2 | 2.04 |
| 4 (14)  | [63 73 1 7 4 88 0 92] | [100 100 0 0 0 100 0 100] | 72.88 | 9.11 | 8.9 | 2.22 |

**Table 3. The optimal simplified model parameter values inferred by the genetic algorithm through global optimisation.** Model parameters are dimensional, taking SI units arising from standard mass action kinetics.

| Parameter | Value (M$^{-1}$s$^{-1}$) | Parameter | Value (s$^{-1}$) | Parameter | Value (Ms$^{-1}$) |
|---|---|---|---|---|---|
| $k_{1c}$ | $4.20 \times 10^{-5}$ | $k_{-1c}$ | $8.15 \times 10^{-5}$ | $\alpha$ | $6.29 \times 10^{-1}$ |
| $k_{1f}$ | $2.64 \times 10^{-5}$ | $k_{-1f}$ | $8.15 \times 10^{-5}$ | - | - |
| $k_{2c}$ | $4.20 \times 10^{-5}$ | $k_{-2c}$ | $8.15 \times 10^{-5}$ | - | - |
| $k_{2f}$ | $2.64 \times 10^{-5}$ | $k_{-2f}$ | $8.15 \times 10^{-5}$ | - | - |
| - | - | $\beta_P$ | $1.69 \times 10^{-3}$ | - | - |
| - | - | $\delta_D$ | $1.66 \times 10^{-4}$ | - | - |
| - | - | $\delta_X$ | $3.31 \times 10^{-3}$ | - | - |

angular metric for the optimal Gillespie simulations for 255 of the 256 possible BLADE circuits has a mean of 1.80˚ and standard deviation of 1.24˚ implying that the majority of circuits perform very well, with an adapted angular metric score close to the optimal 0˚ (S2 Fig). Note that 255 circuits were simulated since the [STOP STOP STOP STOP] circuit is a trivial case that prescribes no functionality, and was unsurprisingly not tested experimentally. In addition, this circuit causes a division by zero in calculating its predicted adapted angular metric score and is therefore not considered in our analyses. The distribution of simulations specifically for those circuits that have been tested experimentally has a mean of 1.90 and a standard deviation of 1.30 (S3 Fig). The distribution of the error between these simulations and the corresponding experimental data is skewed towards zero, with a mean of 1.28 and standard deviation of 1.19 (Fig 6). This implies that Gillespie simulations of the experimentally tested circuits are generally very accurate, with over 50% within 1˚ of the data and 85% within 2˚ of the data. Finally, we examine the performance of 142 of the 143 BLADE circuits that were not tested experimentally (Fig 7). The distribution of model predictions for these circuits has a mean of 1.72 and a

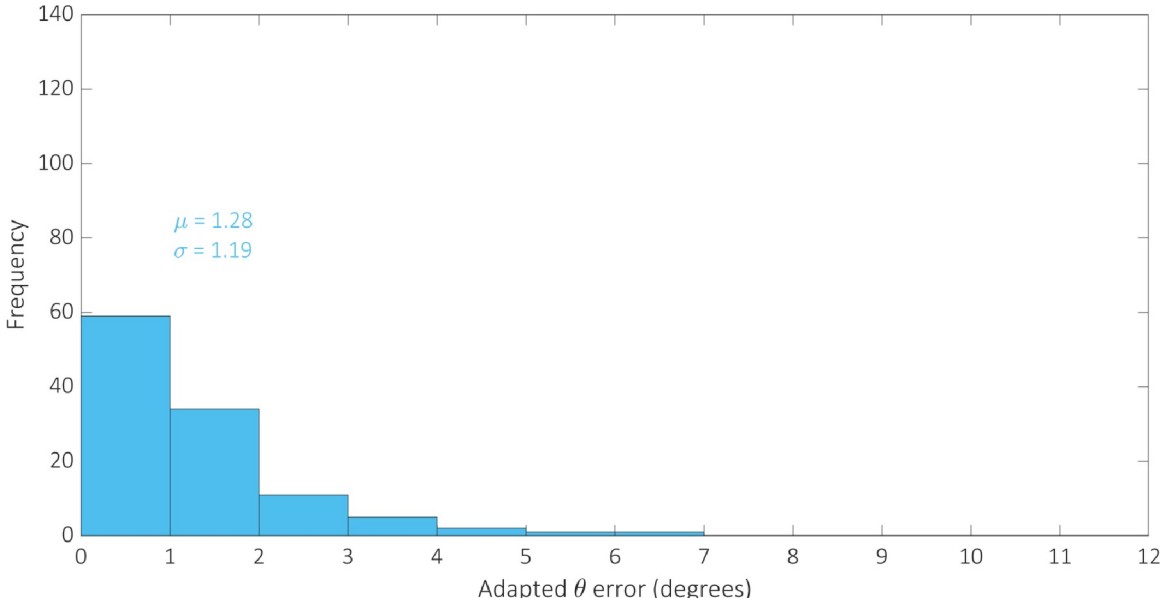

**Fig 6. Histogram showing the distribution of the error between Gillespie model simulations of the adapted metric scores for the 113 BLADE circuits tested experimentally in [8], and the corresponding experimental data.** The mean and standard deviation are denoted by μ and σ respectively. The majority of circuits deviate from their corresponding experimental performance by up to 2˚ when simulated using our model.

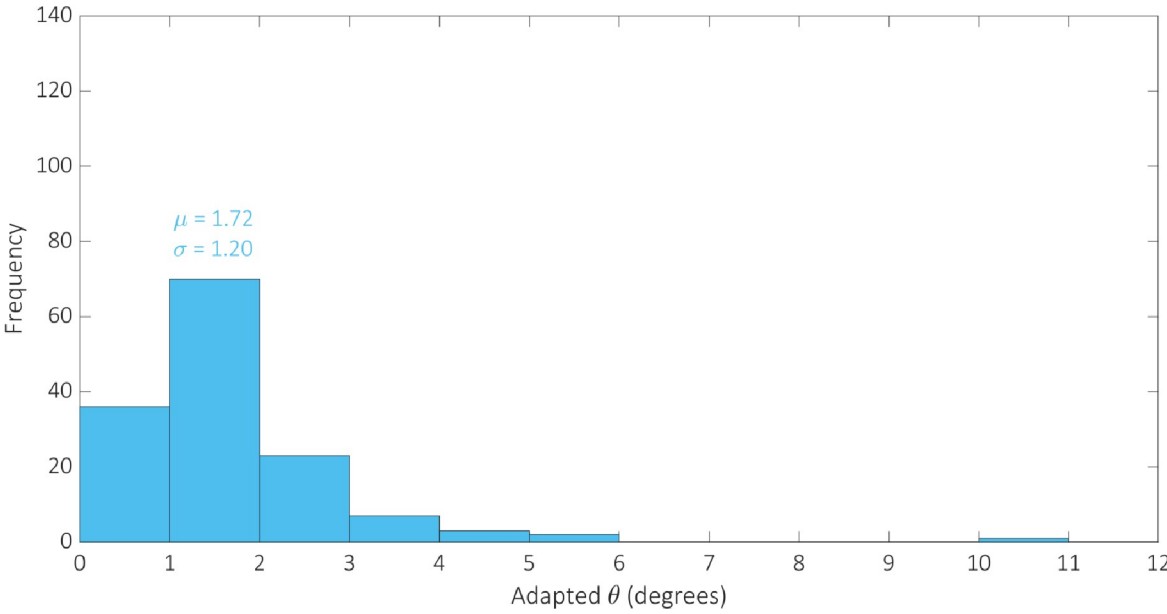

**Fig 7. Histogram showing the distribution of Gillespie simulations of the adapted metric scores for 142 of the 143 BLADE circuits that were not tested experimentally in [8].** The mean and standard deviation are denoted by μ and σ respectively. The majority of BLADE circuits not yet tested experimentally are predicted to perform strongly, with an angular metric score less than 2˚.

standard deviation of 1.20 which implies that, in practice, these circuits would also be expected to perform to the same standards, with adapted angular metric scores close to the optimal 0˚ and a relatively low spread.

To further illustrate the data fitting, model calibration and predictions described by the aforementioned distributions, we plot four Gillespie model simulations relating to circuits that were tested experimentally (including those used to train the model) against their corresponding experimental data, and four simulations relating to circuits that were not tested experimentally (Fig 8). There is minimal error regarding the data fitting circuits, which increases only slightly for the two other calibration circuits selected from the set of 111 on the basis that they consist of each of the four outputs. Predictions also appear to be low scoring for those circuits not tested experimentally, which were also selected on the basis that they consist of each of the four outputs.

## The composition of circuit outputs affects their performance

Although the distribution of the model simulations across all circuits reveals that circuit performance is generally very strong, there are a small number of circuits whose performance is relatively poor. To investigate the cause of this poor performance, with respect to the experimentally tested circuits, we identify the worst performing circuits by plotting the simulated adapted metric score for each circuit in ascending order (Fig 9). There are 13 circuits whose scores are more than one standard deviation greater than the mean ($\theta > \mu + \sigma$) which we classify as exhibiting poor performance. These circuits are composed of the four output genes in the following proportions: GFPmCherry 4%, GFP 27%, mCherry 25%, STOP 44%. By comparison, the composition of the 100 circuits whose scores are less than one standard deviation more than the mean ($\theta < \mu + \sigma$) take the following proportions: GFPmCherry 24%, GFP 25%, mCherry 27%, STOP 24%. The increase in the percentage of STOP sequences and the decrease in the percentage of GFPmCherry sequences is significant for the circuits that perform poorly.

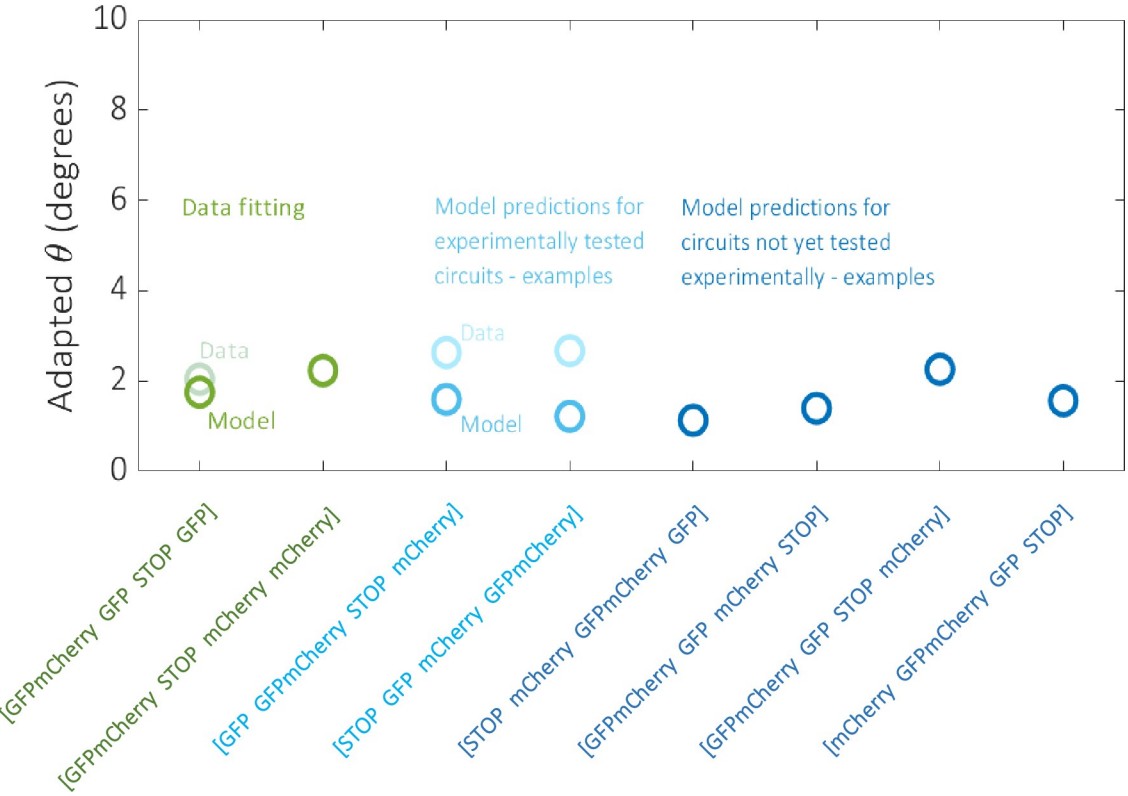

**Fig 8. Data fitting results and example model predictions.** The model was fit to the processed experimental data for the two circuits denoted in green. Two examples of model predictions against data for the remaining 111 circuits tested experimentally are denoted in light blue. The model was also used to predict the performance of the 142 non-trivial circuits that were not tested experimentally, four examples of which are denoted in dark blue. Example circuits were selected on the basis that they consist of each of the four outputs and therefore have the potential for similar dynamical responses.

This trend can also be identified with respect to the circuits that have not been experimentally tested. In this case, there are 14 circuits for which $\theta > \mu + \sigma$ (Fig 10) with the following compositions: GFPmCherry 7%, GFP 16%, mCherry 27%, STOP 50%. The remaining 128 circuits for which $\theta < \mu + \sigma$ have the following compositions: GFPmCherry 30%, GFP 24%, mCherry 25%, STOP 21%. Hence, it appears that 2-input BLADE circuits perform best when they have fewer STOP sequences and more GFPmCherry sequences. Furthermore, we believe some of the low performance can be attributed to variable expression in different address positions. For instance, we observe lower expression in the last position ($Z_{11}$), probably due to the fact that there are two remaining recombination sites left after excision. We have shown recently that these sites can block transcription.

## Model predictions are validated by new experimental data

Selecting circuits based on their output composition has the potential to limit the range of functions available from the 2-input BLADE platform. Alternatively, it is possible to tune model parameters experimentally in order to improve circuit performance. The data in [8] was generated based on a maximal reporter:recombinase expression ratio, hence we are unable to directly compare data relating to increased recombinase activity to our existing data. However, we are able to reduce this ratio and observe how the performance of the circuits is

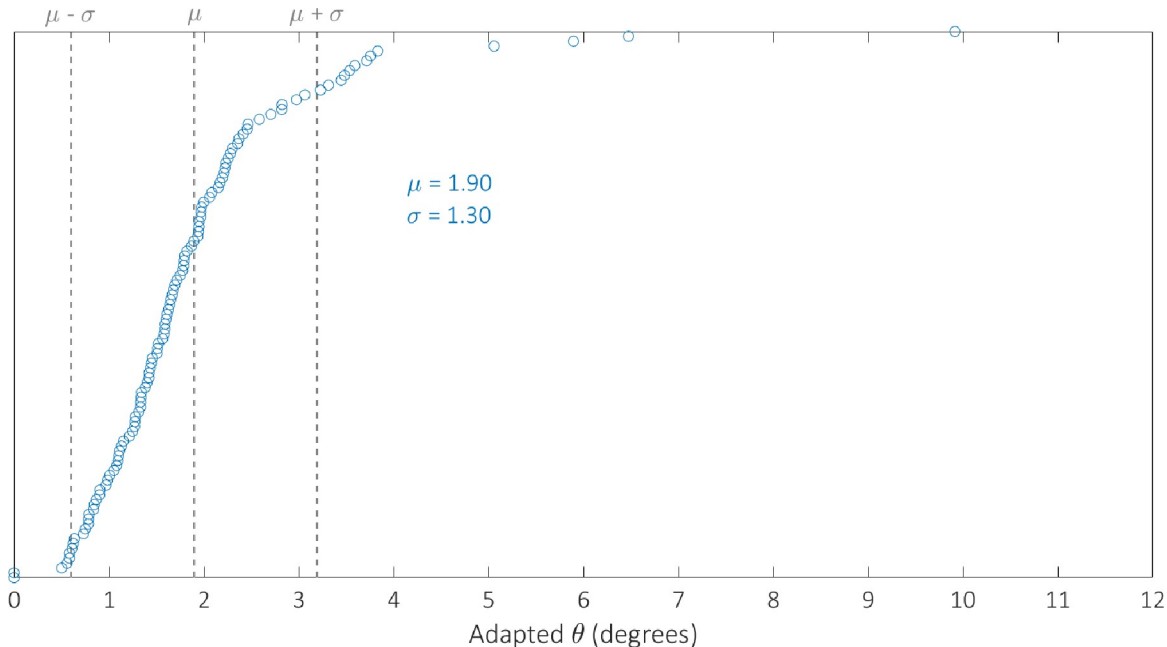

**Fig 9. Model simulations of the performance of the 113 circuits tested experimentally, sorted in ascending order with respect to angular metric score.** The mean and standard deviation are denoted by μ and σ respectively. Dotted lines denote the mean angular metric score (θ = μ) and one standard deviation above and below the mean (θ = μ ± σ). Simulations reveal adapted angular metric scores greater than one standard deviation above the mean for 13 circuits. The remaining 100 circuits all perform within one standard deviation of the mean or better.

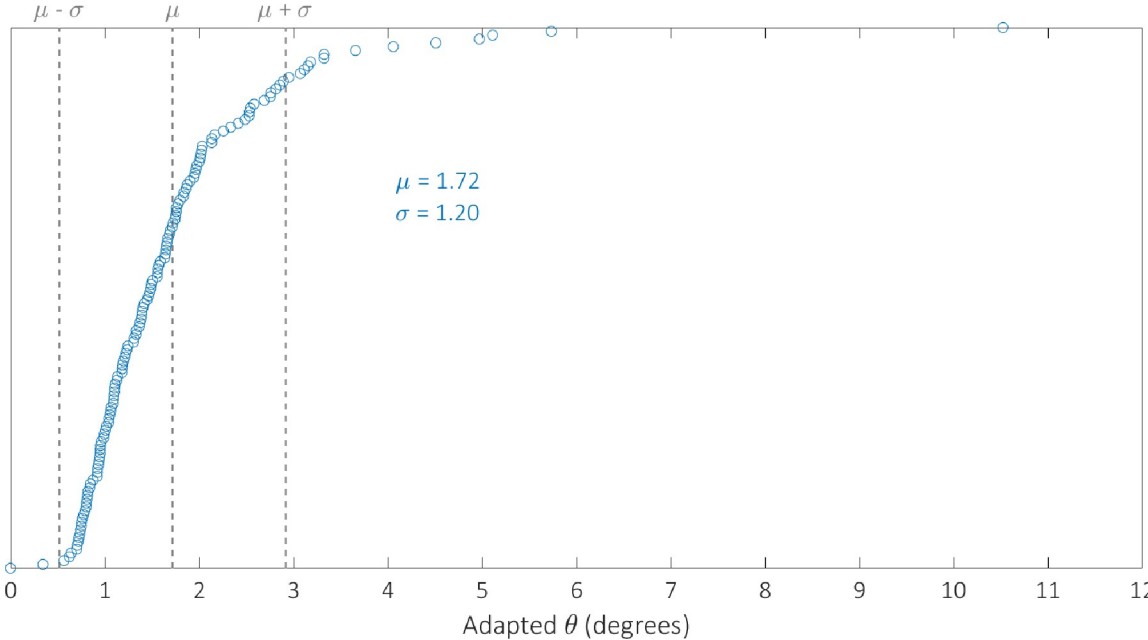

**Fig 10. Model predictions of the performance of the 142 circuits not tested experimentally, sorted in ascending order with respect to angular metric score.** The mean and standard deviation are denoted by μ and σ respectively. Dotted lines denote the mean angular metric score (θ = μ) and one standard deviation above and below the mean (θ = μ ± σ). Simulations reveal adapted angular metric scores greater than one standard deviation above the mean for 14 circuits. The remaining 128 circuits all perform within one standard deviation of the mean or better.

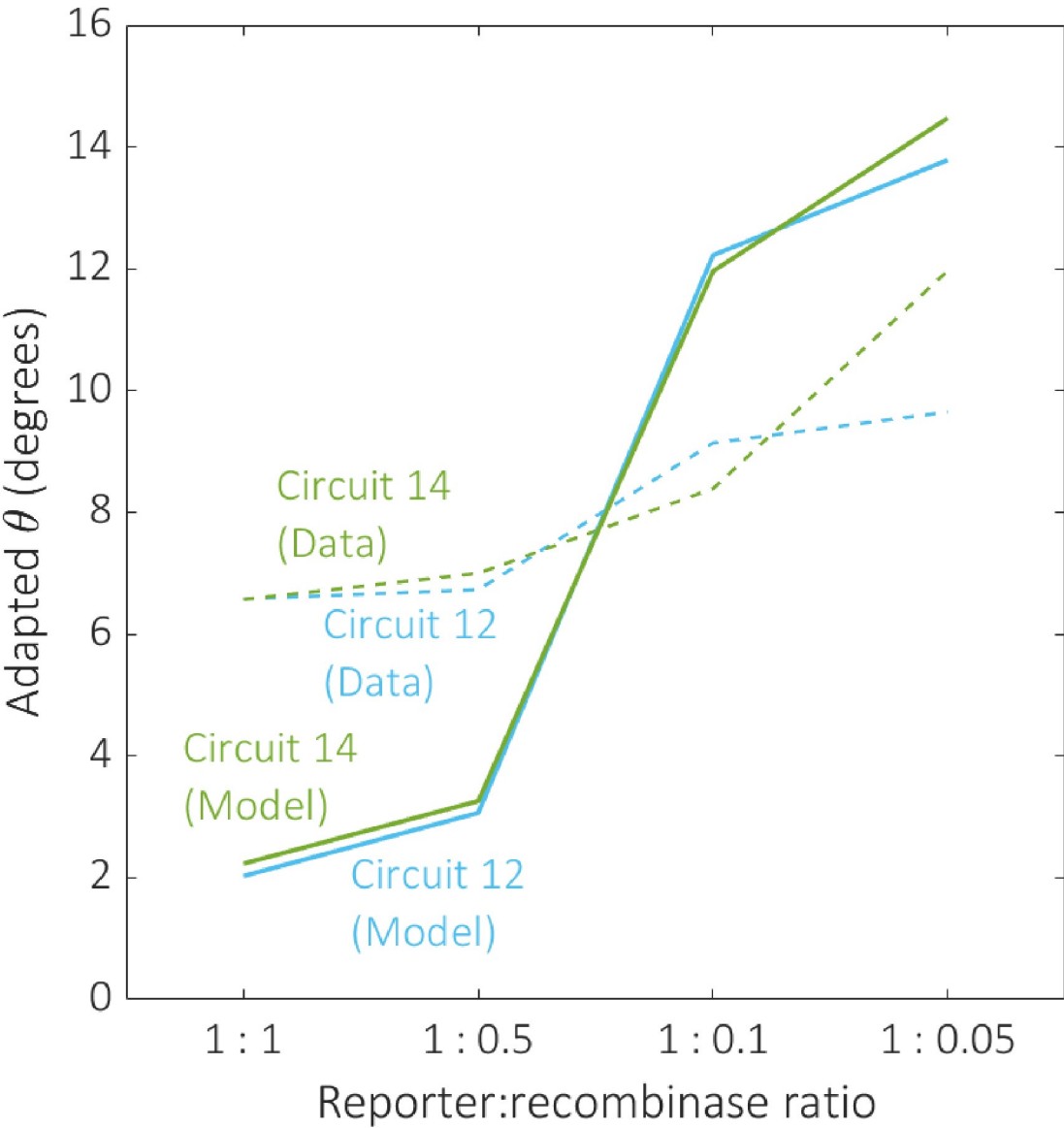

**Fig 11. The effect of decreasing recombinase expression on circuit performance.** The effect of reducing the reporter: recombinase ratio on performance was examined for the same two circuits selected for data-fitting, circuits 12 (blue) and 14 (green). Dotted and solid plot lines correspond to experimental data and predictive model simulations respectively. The model predictions align qualitatively with our data, demonstrating a decline in circuit performance as recombinase expression decreases.

affected. Our new data records the percentage of cells ON subject to four reporter:recombinase ratios (1:1, 1:0.5, 1:0.1, 1:0.05), with respect to the same two circuits that were used for data-fitting (Table 2). By dividing the model parameter describing recombinase expression, α, by the same scaling factors, we are able to predict the effect on circuit performance by virtue of our adapted angular metric (Fig 11). The 1:1 ratio represents the same experimental conditions as the data recorded in [8]. The effect of decreasing the recombinase activity in the system is an overall loss of performance, causing an increase in the adapted angular metric scores for both circuits. This relationship is predicted by our model, confirming its usefulness as a tool for simulating the performance of 2-input BLADE circuits.

## Conclusions

Mathematical models are useful tools for analysing the performance of synthetic biological systems. We have developed the first mechanistic model of the BLADE platform based on our existing model of the tyrosine recombinase-based FLEx switch. The full mechanistic model was simplified in order to produce computationally feasible ensembles of stochastic simulations, using the Gillespie algorithm. The stochastic model was optimised against adapted angular metric data relating to two of the BLADE circuits tested experimentally. Our calibrated stochastic model accurately predicted the adapted angular metric performance of the remaining 111 circuits tested experimentally, as evidenced by the distribution of the error associated with these simulations. Model simulations of the 143 circuits that were not tested experimentally revealed comparable performance levels to those that were tested. Examination of the worst performing circuits revealed that they contain a significantly high/low percentage of STOP/GFPmCherry sequences compared with the vast majority of circuits that perform well. The model was also used to predict the effect of decreasing the expression of recombinases relative to the reporter used to quantify the performance of each circuit. We generated new data for the two circuits used to optimise the model which showed a close match to our predictions, confirming that we have developed a tool capable of accurately simulating any BLADE circuit.

## Materials and methods

### Experimental procedure

Human embryonic kidney cells line (HEK293FT) was maintained in a humidified incubator at 37°C and 5% $CO_2$ with DMEM medium (Corning) with 5% heat-inactivated foetal bovine serum (Life Technologies), 50 UI/mL penicillin, 50 μg/mL streptomycin (Corning), 2 mM glutamine (Corning) and 1 mM sodium pyruvate (Lonza). Polyethylenimine (PEI) mediated transient transfection protocol was utilised to introduce DNA into cells. One day before transfection, HEK 293FT cells were lifted in suspension. Cell density was adjusted to 200,000 cells/ml. 96 well plates were plated with 100 μl cell suspension per well. On the day of transfection, Sterile PEI stock (0.323 g/L), which were stored at -80°C, was warmed at room temperature before use. A total of 400 ng DNA was prepared for each experiment group (4 transfection replicates). The weight of reporter plasmids (a quarter of the total DNA weight) was kept consistent among experiments, while the reporter to recombinase ratio varied (1:1, 1:0.5, 1:0.1, and 1:0.05). Besides reporter and recombinase, the transfection DNA mixture also included a quarter transfection marker plasmid (pCAG-tagBFP), and a blank plasmid (pCAG-FALSE) which was added so the total DNA weight was 400 ng. Sodium chloride (NaCl) solution (0.15 M) was used to bring the volume up to 20 μl, then another 20 μl of PEI: NaCl (PEI:0.15M NaCl = 1:16) mixture was added, resulting in 40 μl end transfection mixtures. The final transfection mixtures were incubated at room temperature before added slowly to cells in HEK293FT cells in 96 well plates (10 μl/well) to make 3 transfection replicates for each experiment group. HEK293FT cells were incubated with transfection mixtures at 37°C and 5% $CO_2$ for 48 hours before being analyzed by a Life Technologies Attune Nxt 4-laser acoustic focusing flow cytometer using laser configurations for detection of EGFP (488 nm laser, 510/10 emission filter), mCherry (561 nm, 620/15 emission filter), and tagBFP (405 nm laser, 440/50 emission filter) Flow cytometry data was analyzed using FlowJo (Tree Star). A live population gate was applied according to forward scatter and side scatter to eliminate dead cells and other debris. Of the live cell population, the top 0–0.1% wild type cells on the transfection marker (tagBFP) fluorescence channel was gated as the transfection marker for positive cells. Reporter mean fluorescent intensity (MFI) of the transfected population was collected

from EGFP and mCherry channel and averaged among transfection replicates for subsequent processing. Transient transfection is the most reasonable way to evaluate >100 synthetic circuits in human cells; see [8] for full details of the experimental implementation of the BLADE platform and its applications.

## Stochastic simulations using the Gillespie algorithm

The Gillespie algorithm performs stochastic simulations of systems of biochemical reactions, without the need to solve the stochastic master equation. This approach is considered to have greater physical validity than traditional deterministic methods, since these systems are typically observed with respect to populations of cells rather than on a single-cell level. This accounts for the randomness that governs the nature in which cellular interactions occur. The algorithm is initialised with five key parameters: the total number of reactions that occur in the system, the total number of molecular species, the total number of molecules, the initial number of molecules associated with each species at time zero, and the maximum number of reactions to be simulated. In our case, the following values were chosen:

- Total number of reactions: 17

- Total number of molecular species: 10

- Total number of molecules: 1000

- Initial number of molecules: Initially all molecular species have 0 molecules, except for $Z_{00}$ which has 1000

- Maximum number of reactions to be simulated: 10000

The reaction rate constants and molecular reactant combinations associated with each reaction provide a set of propensities that resemble the equivalent deterministic terms. The time step between each reaction is calculated using a randomly generated number and the sum of the initial propensities. A second randomly generated number is used along with the propensities to determine which specific reaction will take place at each time step. Initially, the species with the greatest number of molecules will have the greatest propensity and therefore reactions involving this species will have the greatest probability of occurring. The simulation then progresses through successive randomly generated time intervals. At each time point, the propensities are updated based on the new distribution of molecules and the next reaction is chosen probabilistically. As a result, the reaction that occurs at each time step is not guaranteed to be that which is most likely to occur, and hence the simulation evolves in a stochastic manner. The simulation terminates when the reaction counter reaches its predefined maximum value. We employed MATLAB R2018a to run Gillespie algorithm simulations, and have uploaded this MATLAB code along with an SBML version of our simplified model to GitHub.

## Parameter inference using a genetic algorithm

We employed the built-in GA function in MATLAB R2018a for parameter inference purposes. The MATLAB code used to generate our results can be found on GitHub. The GA proceeds by initially generating a population of solutions at random within the predefined parameter space. The initial solutions are then scored based on their fitness. This is typically calculated by virtue of an error function that determines how well each solution is able to match the relevant experimental data. The best solutions are selected as parents that will produce the best offspring to populate the next generation of solutions. Parents produce offspring through crossover, whereby a random place in their binary genotype is selected and the information beyond

that point is swapped over. Mutation is also incorporated, whereby single point alterations in the offspring's genotype are imposed to increase diversity within the population. This procedure is repeated indefinitely until a predefined termination criterion is reached; typically, the scores are unchanged over a specified number of generations, the algorithm reaches a specified number of generations, or a specified solution score is established. Parallelisation of the MATLAB code enabled us to run the GA with a large search population for many generations; this significantly increases of likelihood of establishing the global optimal solution.

Given the lack of available data regarding the relevant reaction rate constants in the literature, we require a parameter space large enough to locate optimal solutions, but not so big that convergence timescales become impractical. A large parameter space increases the time taken to produce the next generation of solutions however, this can also decrease the overall number of generations required for convergence, since more solutions are inspected in each case. Large parameter spaces also increase the risk of incurring problems with stiff model simulations that may cause the GA to fail, hence multiple trials are often required to determine effective performance criteria. Hence, the search interval imposed on all model parameters is $[10^{-5}, 10^5]$. We ran the GA over 1000 generations with a population size of 100 in order to maximise the likelihood of convergence and the identification of the global optimum solution.

Applying the GA to stochastic Gillespie algorithm simulations is challenging because, unlike a deterministic model, stochastic simulations do not provide identical solutions for the same parameterisation. We overcame this by using an objective function, $E$, that calculates the mean absolute error between the experimental data and the mean of the ensemble simulations. The standard deviation is included in the objective function so that the variance/spread is also minimised i.e.

$$E = |\bar{\theta}_1 - d_1| + \sigma_1 + |\bar{\theta}_2 - d_2| + \sigma_2,$$

where $\bar{\theta}_1$ and $\bar{\theta}_2$ are the mean adapted angular metric scores for an ensemble of 100 Gillespie model simulations of circuits 3 and 4 (Table 2) respectively; $d_1$ and $d_2$ are experimental data for the adapted angular metric scores for circuits 3 and 4 respectively; $\sigma_1$ and $\sigma_2$ are the standard deviations of the ensemble simulations for circuits 3 and 4 respectively.

## Supporting information

**S1 Fig. Reaction network (full model).** Schematic diagram of the tyrosine recombinase-mediated 2-input BLADE platform. Tyrosine recombinases Cre (red circles) and Flp (blue circles) are expressed constitutively at the same rate, $\alpha$. LoxP sites are depicted as red triangles with different shades used to illustrate the result of recombination events. FRT sites are depicted as blue triangles with different shades used to illustrate the result of recombination events. White boxes depict each of the four BLADE addresses. Black arrows depict DNA:protein binding reactions comprising stable excision events. The rate of degradation of recombinase protein is denoted by $\beta_p$. The rate of a Cre/Flp monomer binding reversibly to free loxP/FRT sites is denoted by $k_1$, $k_{-1}$; due to the cooperativity of monomer binding, we denote the rate of a Cre/Flp monomer binding reversibly to an occupied loxP/FRT site by $k_2$, $k_{-2}$. The rate of Holliday junction formation is denoted by $k_3$, $k_{-3}$. Each of the five Holliday junction strand exchanges are denoted by $k_4$, $k_{-4}$, $k_5$, $k_{-5}$, $k_6$, $k_{-6}$, $k_7$, $k_{-7}$, respectively. The rate of dilution of excised DNA due to cell division is denoted by $\delta$. The empty set symbol is used to depict expression and degradation reactions.
(PDF)

**S2 Fig. Histogram.** Histogram showing the distribution of Gillespie simulations of the adapted metric scores for 255 of the 256 possible BLADE circuits. The mean and standard deviation

are denoted by μ and σ respectively. The majority of circuits perform within 2° of optimal performance (0°). Gillespie simulations performed using the optimal parameter set identified through global optimisation.
(PDF)

**S3 Fig. Histogram.** Histogram showing the distribution of Gillespie model simulations of the adapted metric scores for the 113 BLADE circuits tested experimentally in [8]. The mean and standard deviation are denoted by μ and σ respectively. The majority of circuits perform within 2° of optimal performance (0°). Gillespie simulations performed using the optimal parameter set identified through global optimisation.
(PDF)

**S4 Fig. Histograms.** Histograms showing the distributions of final population GA solutions for a separate run (blue). The orange bars are not distributions, but indicate the bin in which the corresponding optimal parameter value is located.
(PDF)

**S1 Table. Biochemical equations.** Biochemical equations describing the full reaction network of the 2-input BLADE platform (S1 Fig). Reactions are paired for the two separate Flp-mediated excision events for the sake of brevity i.e. one set of equations for both $f_1$ and $f_2$ (right column), since these two events consist of identical reactions. The left column lists the equivalent reactions that correspond to both Cre-mediated excision events, $c_1$ and $c_2$, however the middle column lists those reactions unique to $c_1$, due to the fact that $c_1$ consists of more reactions than $c_2$ as well as $f_1$ and $f_2$. Cre and Flp that are expressed constitutively and mediate the four excision events are denoted by C and F, respectively. Cre and Flp monomers bind to DNA attachment sites sequentially until two monomers occupy each site; $C_{1,1}$,$F_{1,1}$ and $C_{2,0}$,$F_{2,0}$ denote one monomer bound to each site and two monomers bound to one site, respectively. DNA is denoted by D with a subscript corresponding to one of the four DNA states ($D_{00}$, $D_{10}$, $D_{01}$, $D_{11}$) or four excised DNA states ($D_{00X}$, $D_{10X}$, $D_{01X}$, $D_{11X}$), and with a superscript corresponding to one of the four recombination events ($c_1$, $c_2$, $f_1$, $f_2$). Each of the five strand exchanges that comprise a Holliday junction is denoted by the corresponding numbered H.
(PDF)

**S2 Table. Model ODEs.** Mechanistic model ODEs derived from biochemical equations through the application of mass action kinetics.
(PDF)

**S1 Text. Model simulations (full model).**
(PDF)

## Author Contributions

**Conceptualization:** Benjamin H. Weinberg, Wilson W. Wong, Declan G. Bates.

**Data curation:** Chloe Ding, Benjamin H. Weinberg.

**Formal analysis:** Jack E. Bowyer.

**Funding acquisition:** Wilson W. Wong, Declan G. Bates.

**Investigation:** Jack E. Bowyer.

**Methodology:** Jack E. Bowyer, Wilson W. Wong, Declan G. Bates.

**Project administration:** Wilson W. Wong, Declan G. Bates.

**Resources:** Benjamin H. Weinberg, Wilson W. Wong, Declan G. Bates.

**Software:** Jack E. Bowyer.

**Supervision:** Benjamin H. Weinberg, Wilson W. Wong, Declan G. Bates.

**Validation:** Jack E. Bowyer.

**Visualization:** Jack E. Bowyer.

**Writing – original draft:** Jack E. Bowyer.

**Writing – review & editing:** Jack E. Bowyer, Chloe Ding, Wilson W. Wong, Declan G. Bates.

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
