## [Decision Letter · Decision Letter 0]

18 Jun 2020

Dear Prof. Bates,

Thank you very much for submitting your manuscript "A mechanistic model of the BLADE platform predicts performance characteristics of 256 different synthetic DNA recombination circuits" for consideration at PLOS Computational Biology.

As with all papers reviewed by the journal, your manuscript was reviewed by members of the editorial board and by several independent reviewers. In light of the reviews (below this email), we would like to invite the resubmission of a significantly-revised version that takes into account the reviewers' comments.

We cannot make any decision about publication until we have seen the revised manuscript and your response to the reviewers' comments. Your revised manuscript is also likely to be sent to reviewers for further evaluation.

[2] Two versions of the revised manuscript: one with either highlights or tracked changes denoting where the text has been changed; the other a clean version (uploaded as the manuscript file). All of the reviewers have requested clarification about the models used; in particular, their parameterization. I strongly encourage the authors to follow the request of Reviewer 2 that the models be provided in electronic format to ensure reproducibility of the results. 

Sincerely,

James R. Faeder

Associate Editor

PLOS Computational Biology

Mark Alber

Deputy Editor

PLOS Computational Biology

Reviewer's Responses to Questions

**Comments to the Authors:**

Reviewer #1: Bowyer et al

A mechanistic model of the BLADE platform predicts performance…

1) Applications

To be relevant to a broad audience, including the readers of PLOS CompBio, it would be helpful to motivate the need for the BLADE platform with two or three compelling examples of how it could or would be used.

2) Complex vs Simplified models

It seems the simplified model (Fig 6) was the basis for the stochastic simulations and evaluation of the model-vs-experiments. The full complex model (Fig 2) is useful to show for completeness; but please consider moving it to Supplementary information. As a general reader, I wasted a lot of time trying to decipher its meaning before discovering Fig 6, which is more transparent.

3) Practical issues

In the context of applications, how important is it that the cells be readily transfectable? Are the anticipated applications for cell lines rather than primary cells? How sensitive is the output to the 48h post-transfection period and the selection of the top 0.1 percent of transfection positive cells for quantitative analysis. I realize these are not central points to the paper. However, if you wish to engage the broad PLOS CompBio audience in potential real-world applications, I would recommend such issues at least be discussed.

p.10 Equations 1-4 need to be numbered

Reviewer #2: The paper presents results from a model fo the Boolean logic and arithmetic through DNA excision (BLADE) genetic design platform. BLADE is a genetic design technique for mammalian cells that uses Cre and Flp-mediated DNA incision to turn on gates by inverting DNA to activate a gene or removing a terminator to activate a gene. Using this technique, one can construct combinational genetic logic gates. One important point that was not emphasized in the paper is that these are not exactly combinational logic gates in that they have state. Once the gate has changed state, they are typically not able to change state again with a change inputs as required to be a true combinational logic gate. The authors develop both an ODE and stochastic model and compare their results with experimental results for 111 circuits. They also make predictions about performance for an additional 143 circuits that have yet to be experimentally verified.

While the paper is generally well written and the model appears to be useful, a number of important details are omitted making it difficult to fully judge the model and reproduce any of the results. While a detailed ODE model is presented in the supplemental, the parameters are not provided. Parameters are only provided for a simplified model that I could not find in the paper or the supplemental. There is a simplified ODE model in the paper, but it refers to a k3 and k-3 parameters that I could not find. The stochastic model is even less detailed. There is a biochemical reaction model in the supplemental, but it does not appear that that is what they simulated using the Gillespie algorithm. The authors also do not mention what software they used for their analysis. All of these problems could be addressed by simply providing the model in electronic form (preferably in SBML), as well as instructions for running the model in the software they used. If they wrote their own software, this should also be released using github or similar. Until this information is released it is impossible to make a complete assessment of their results.

Reviewer #3: Bower et al. developed mechanistic model based on biochemical mass action kinetics to describe Flp- and Cre- induced DNA recombination in 256 circuits constructed out of a 2-input and 2-output BLADE platform. The authors use published and new data to estimate parameters in their mechanistic model and test model predictions. The mechanistic model is able to describe fluorescence data obtained from several circuits well and the trained models were used to generate predictions in other circuits; some of which were investigated in published experiments. The main conclusion from the study is that the circuit performance decreases with increasing number of STOP sequences and low number of recombinase molecules. I think the authors address an important problem of developing mechanistic computational models to describe Boolean synthetic biology circuits. They carry out a detailed investigation of their models to arrive at somewhat straigthforward conclusions. In addition, there are few major shortcomings in the present work: 1) It is not clear how well the model parameters can be estimated. 2) The models and the estimated parameters were not used to glean any mechanisms that underlie DNA recombination. In addition, the text should be revised at several places to better describe their system, approach, and results. I provide further details regarding these issues below.

Major comments

1. The kinetic rates used in the model are estimated by minimizing a cost function using genetic algorithm. It is not clear how well the parameters described the data, for example, whether there are parameters with large confidence intervals or there are non-identifiable parameters. In the absence of that it makes it difficult to assess if the models are well parameterized. It also makes it difficult to investigate potential sources for the large differences between the model prediction and the data (e.g., Fig. 9 and Fig. 12). Though the qualitative trends somewhat agree in Fig. 12, for a model that is fitted well with the training data, it is unclear why the gap between the model prediction and the data will be this large. Perhaps, this disagreement points to identifiability issues in the current parameterization and/or the existence of alternate models?

2. The authors make no attempt to extract any mechanism from the estimated parameters or by generating results under a condition that can be informative about the underlying kinetics. The two main conclusions appear to be somewhat straightforward. For example, the conclusion that increasing number of STOP sequences lead to circuit lower performance probably can be inferred by systematic analysis of the published fluorescent data. As for the other conclusion, the lower performance of at the low copy numbers of recombinases could possibly arise due to the increase in the intrinsic noise fluctuations in the chemical kinetics which is kind of expected. Thus, it might be worthwhile to further analyze the models for obtaining non-intuitive/new mechanisms.

3. I was hoping to find some mechanistic reasons behind the low circuit performance in experiments. Does intrinsic noise fluctuation have a role to play here or there are biological variables, not accounted for in the experiments (and in the model), result in the difference between the truth table and a circuit output? The authors also mention co-operativity in recombinase binding, could this co-operativity increase the effect of intrinsic noise fluctuations? It was unclear if this co-operative interaction was treated correctly in their reduced model.

Minor comments

I think it will be helpful to the readers if some parts of the text are revised to explain some of the terms that are used frequently. For example, authors discuss measurements and simulation of in “circuits” using the BLADE pattern. Perhaps, introducing the concept of circuits composed of the BLADE platform will be helpful to a non-expert reader.

**Have all data underlying the figures and results presented in the manuscript been provided?**

Reviewer #1: Yes

Reviewer #2: No: The models should be provided in a standard format such as SBML to ensure reproducibility.

Reviewer #3: Yes

PLOS authors have the option to publish the peer review history of their article (what does this mean?). If published, this will include your full peer review and any attached files.

Reviewer #1: No

Reviewer #2: No

Reviewer #3: No
---

## [Decision Letter · Decision Letter 1]

19 Oct 2020

Dear Prof. Bates,

Thank you very much for submitting your manuscript "A mechanistic model of the BLADE platform predicts performance characteristics of 256 different synthetic DNA recombination circuits" for consideration at PLOS Computational Biology. As with all papers reviewed by the journal, your manuscript was reviewed by members of the editorial board and by several independent reviewers. The reviewers appreciated the attention to an important topic. Based on the reviews, we are planning to accept this manuscript for publication, but we would like to ask that you make a few slight modifications to the manuscript according to the review recommendations. Provided that you address the change requests in your reply, the manuscript will not be sent to the reviewers for further review.

Sincerely,

James R. Faeder

Associate Editor

PLOS Computational Biology

Mark Alber

Deputy Editor

PLOS Computational Biology

[LINK]

Reviewer's Responses to Questions

**Comments to the Authors:**

Reviewer #2: The authors have addressed my concerns with their modeling effort. They have both updated their paper, as well as provided matlab code and an SBML model on github. My only last minor concern is it would be good to add a README file on the github repository to explain the files and how to run the analysis to reproduce their results.

Reviewer #3: The authors have addressed most of my comments. In some cases, the text (main text or SI) does not reflect explanations provided in the response letter. For example, for Fig. S4, the readers might benefit from knowing how the authors use Fig. S4 to interpret "robustness" in their parameter choice. The authors state in the response letter (comment #1),

"The histogram bin containing the optimal parameter value is also added, and it can be seen that the bins match up for almost all parameters. This shows that distinct optimisation runs provide similar results, and hence it is likely that the parameterisation we have identified is one of only a small number of similar parameterisations that provides the model outputs reported here. "

Perhaps a brief sentence in the main text or the figure legend of Fig. S4 would help.

I think the authors also do not include any potential mechanistic explanation (comment #3) in the main text as they provide in the response,

"Some of the low performance can be attributed to variable expression in different slots. For instance, we observe lower expression in the last slot, probably due to the fact that there are two remaining recombination sites left after excision. We have shown recently that these sites can block transcription."

These mechanistic explanations might be beyond the scope of the work as the authors mention in the letter, therefore, I leave it to the authors if they would like to include this explanation in the text.

**Have all data underlying the figures and results presented in the manuscript been provided?**

Reviewer #2: Yes

Reviewer #3: Yes

PLOS authors have the option to publish the peer review history of their article (what does this mean?). If published, this will include your full peer review and any attached files.

Reviewer #2: **Yes: **Chris J. Myers

Reviewer #3: No
---

## [Editor Report · Decision Letter 2]

3 Nov 2020

Dear Prof. Bates,

We are pleased to inform you that your manuscript 'A mechanistic model of the BLADE platform predicts performance characteristics of 256 different synthetic DNA recombination circuits' has been provisionally accepted for publication in PLOS Computational Biology.

Best regards,

James R. Faeder

Associate Editor

PLOS Computational Biology

Mark Alber

Deputy Editor

PLOS Computational Biology

---

## [Editor Report · Acceptance letter]

3 Dec 2020

PCOMPBIOL-D-20-00525R2 

A mechanistic model of the BLADE platform predicts performance characteristics of 256 different synthetic DNA recombination circuits

Dear Dr Bates,

I am pleased to inform you that your manuscript has been formally accepted for publication in PLOS Computational Biology. Your manuscript is now with our production department and you will be notified of the publication date in due course.

With kind regards,

Nicola Davies
